# A Rare Case of Elbow Synovial Cyst with Radial Nerve Compression

**DOI:** 10.3390/diagnostics15020124

**Published:** 2025-01-07

**Authors:** Ting-Hsuan Hsu, Yen-Nung Lin

**Affiliations:** 1Department of Physical Medicine and Rehabilitation, Wan Fang Hospital, Taipei Medical University, Taipei 116, Taiwan; 108229@w.tmu.edu.tw; 2Graduate Institute of Injury Prevention and Control, Taipei Medical University, Taipei 235, Taiwan

**Keywords:** elbow pain, synovial cyst, radial nerve compression, sonography, neuropathy

## Abstract

Elbow synovial cysts are rare and can mimic more frequently encountered disorders such as lateral epicondylitis, presenting diagnostic challenges. This report describes a woman in her mid-40s with persistent pain and weakness in her right forearm due to a synovial cyst compressing the radial nerve at the Arcade of Frohse. Despite initial suspicions of lateral epicondylitis, deeper investigation using sonography confirmed the presence of a compressive synovial cyst. Ultrasound-guided aspiration of the cyst was performed, yielding clear synovial fluid and providing symptomatic relief. Post-procedure imaging showed a significant reduction in cyst size and alleviation of nerve compression. This case highlights the critical role of integrating sonography with clinical evaluations in diagnosing and managing atypical presentations of neuropathic pain and motor weakness. This advanced imaging capability not only effectively facilitated an accurate diagnosis but also enabled a targeted therapeutic intervention, thereby avoiding extensive surgical procedures and reducing the risk of nerve injury.

**Figure 1 diagnostics-15-00124-f001:**
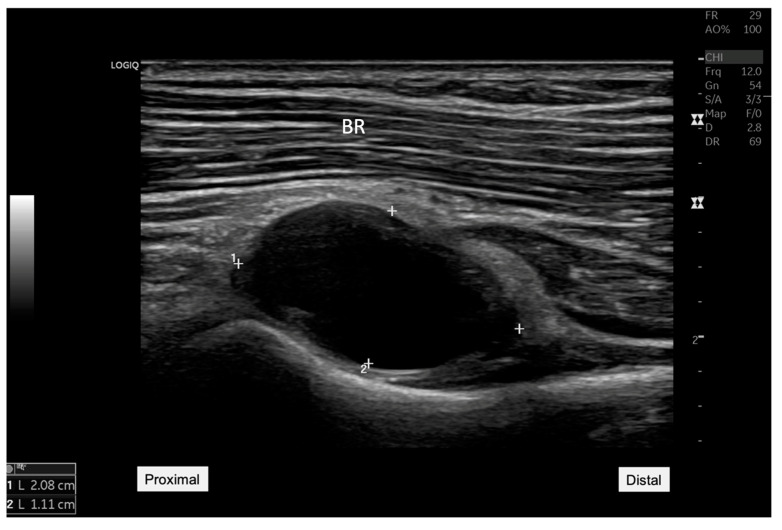
Ultrasound image showing an anechoic cyst measuring 20.8 mm by 11.1 mm located adjacent to the radiocapitellar joint. This finding was critical in reorienting the differential diagnosis toward the possibility of a synovial cyst impacting nearby neural structures. BR: Brachial radialis. A 45-year-old woman presented to our rehabilitation outpatient clinic with a three-week history of persistent pain and weakness in her right elbow and forearm, particularly exacerbated by wrist extension movements. Initial clinical assessments suggested possible lateral epicondylitis [1], but the patient’s lack of response to standard conservative treatments prompted further diagnostic evaluation to explore underlying causes. On physical examination, manual muscle testing revealed significant weakness in right-hand extension involving all five fingers (grade 3 of 5) and wrist extension (grade 3 of 5). However, elbow extension remained strong (grade 5 of 5). Sensory examination revealed no deficits in the right hand or fingers, while the patient reported pain localized to the proximal forearm, notably exacerbated by wrist and finger extension movements. Electrophysiological testing revealed evidence of a conduction block with intact distal compound muscle action potential (CMAP) amplitude, suggesting radial nerve compression. Ultrasound examination was conducted using a GE LOGIQ P8 machine, with a 12-MHz linear transducer, ideal for detailed nerve and superficial soft tissue. Ultrasound imaging revealed a hypoechoic mass, measuring 20.8 mm by 11.1 mm near the radiocapitellar joint (Figure 1). The longitudinal ultrasound view of the radial nerve (Figure 2) illustrates the entrapment of the posterior interosseous nerve (PIN), as it courses through the Arcade of Frohse. Notable compression of the nerve was seen due to an underlying anechoic cyst, which contributes significantly to the PIN’s entrapment at this anatomical site. The cyst’s proximity to the nerve and its effect on the nerve’s appearance and function are critical for diagnosing radial nerve compression. Additionally, nerve tracking to the supinator level also showed distinct hypoechogenicity and swelling of the PIN as it passed through the supinator muscle on the affected side, highlighting the pathological changes consistent with radial nerve compression (Figure 3) [2].

**Figure 2 diagnostics-15-00124-f002:**
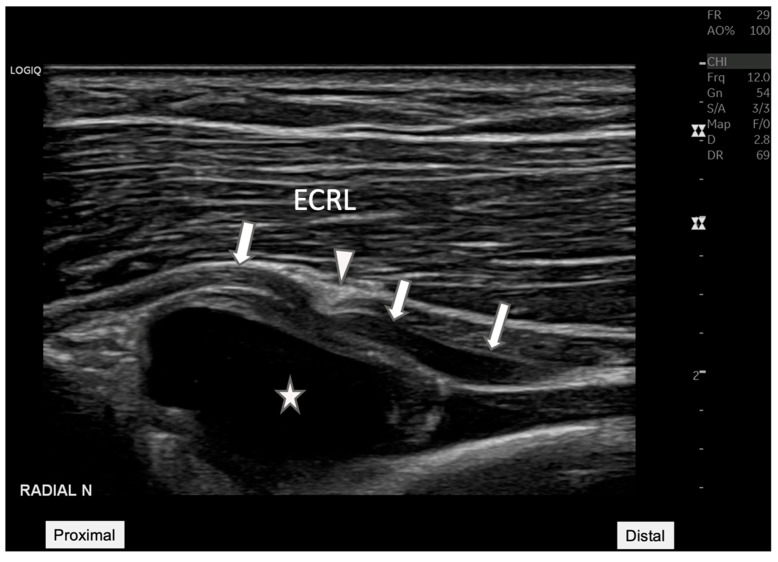
**Radial nerve entrapment at the Arcade of Frohse.** Longitudinal ultrasound view of the posterior interosseous nerve (arrow) depicting the nerve compression at the Arcade of Frohse (arrowhead). The nerve is visibly constricted by an underlying anechoic cyst (star). This image demonstrates the value of ultrasound in visualizing soft tissue structures, providing essential diagnostic insights into the mechanisms of neuropathic pain and potential therapeutic targets. ECRL: Extensor carpi radialis longus.

**Figure 3 diagnostics-15-00124-f003:**
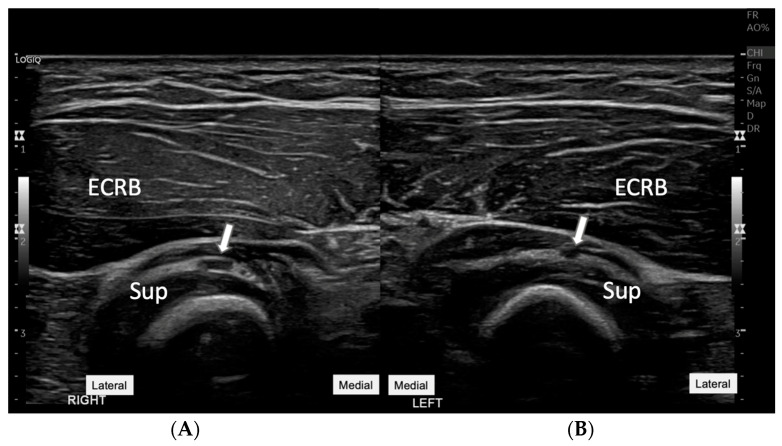
**Comparative ultrasound images of the affected right forearm** (**A**) **with the unaffected left side** (**B**). This figure presents side-by-side images of both the right and left forearms, specifically focusing on the region of the supinator muscles. The arrows on each image point to the PIN as it courses through the supinator muscle (Sup). On the right side (**A**), there is a noticeable swelling of the PIN at the site of the lesion. In contrast, the left side (**B**) shows a normal appearance of the PIN without swelling, serving as a control reference. This comparative view underlines the diagnostic value of ultrasound in assessing asymmetrical nerve conditions and supports clinical findings of neuropathy on the affected side. ECRB: Extensor carpi radialis brevis. An ultrasound-guided aspiration was performed, carefully avoiding the PIN (Figure 4). The precision of the procedure, with the aspiration needle positioned to avoid nerve structures while targeting the cyst, was critical [3]. The aspiration yielded a clear synovial fluid, which not only confirmed the nature of the cyst but also relieved pressure on the PIN. The differential diagnosis for the mass included conditions such as lipoma, rheumatoid nodule, and other inflammatory processes like bursitis or abscess formation, which typically present with a different echogenicity pattern, helping to rule them out in this case [4,5].

**Figure 4 diagnostics-15-00124-f004:**
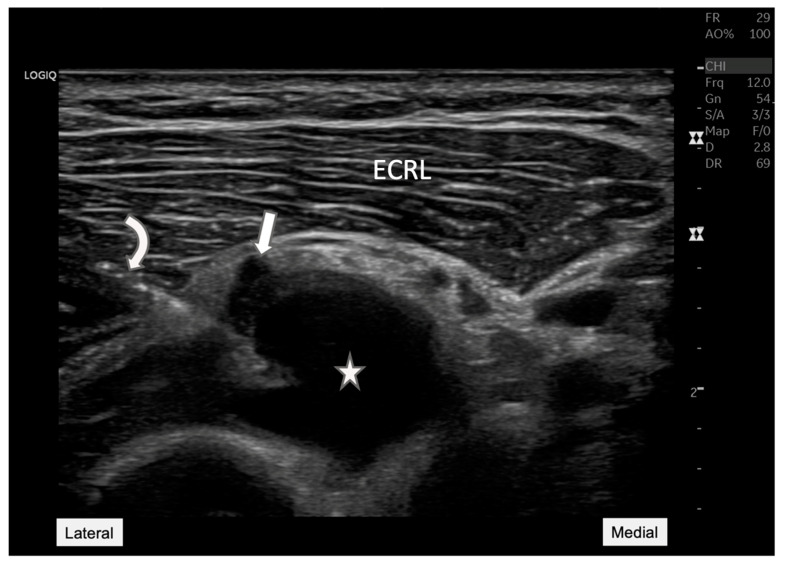
**Ultrasound-guided aspiration.** This ultrasound image illustrates a transverse view of the right forearm to target the synovial cyst (star). The PIN (straight arrow), which was meticulously avoided by the aspiration needle (curved arrow). This image demonstrates the critical importance of ultrasound in guiding minimally invasive procedures, ensuring both the safety and accuracy of the intervention by avoiding vital structures like nerves while targeting pathological lesions. This technique minimizes the risk of complications and maximizes therapeutic efficacy. ECRL: Extensor carpi radialis longus. Following the ultrasound-guided aspiration of the synovial cyst, the patient experienced immediate relief from the previously reported symptoms of pain in her right forearm [6]. This immediate improvement confirmed both the accuracy of the diagnosis and the efficacy of the procedure, significantly enhancing the patient’s quality of life. One and a half months following an ultrasound-guided intervention, the patient returned for a follow-up evaluation. The post-procedure imaging (Figure 5) was highly encouraging, demonstrating a significant reduction in the cyst’s size and substantial alleviation of nerve compression. Clinically, these improvements correlated directly with the patient’s own experiences; she reported a marked decrease in pain and a restoration of forearm strength. This outcome not only highlights the effectiveness of the intervention but also underscores the critical role of precise diagnostic and therapeutic techniques in managing complex cases of neuropathic pain. The successful resolution of symptoms following the procedure reaffirms the utility of ultrasound-guided aspiration in treating synovial cysts that cause compressive neuropathies, offering patients significant relief and a return to normal function.

**Figure 5 diagnostics-15-00124-f005:**
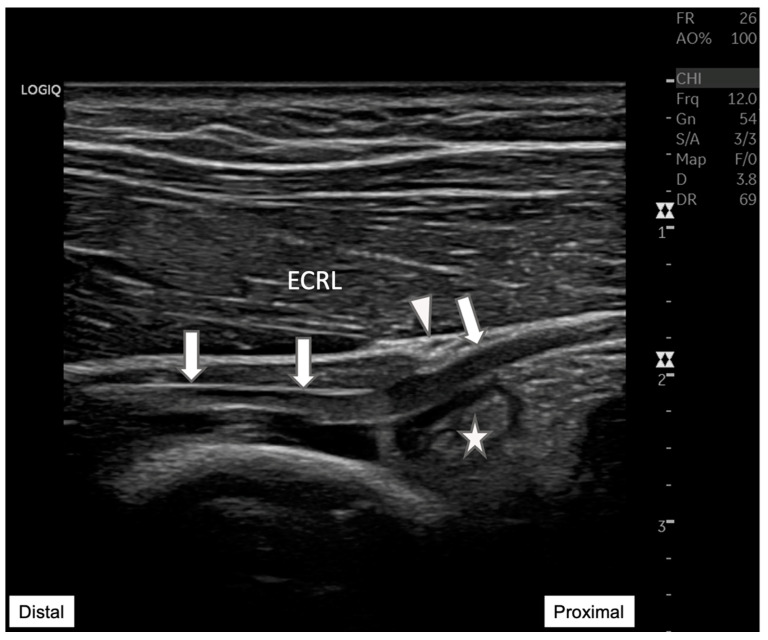
**Follow-up ultrasound evaluation of the radial nerve in the forearm.** This ultrasound image provides a clear view of the radial nerve (arrow) after an aspiration procedure targeting a synovial cyst (star). PIN at the Arcade of Frohse (arrowhead), where prior significant compression has been notably alleviated following the intervention and the path of the nerve, which had a reduced impingement. The previously larger cyst, which has been markedly reduced in size due to successful aspiration. This image demonstrates the effectiveness of ultrasound-guided interventions in managing compressive neuropathies. ECRL: Extensor carpi radialis longus. Previous reports of PIN compression caused by synovial cysts at the Arcade of Frohse are limited. Boushabi et al. [7] highlighted a case diagnosed using MRI, where surgical excision successfully relieved compression and restored motor function without recurrence. Another study [8] involving eight patients with PIN palsy identified synovial cysts as the cause in most cases. Diagnosis The diagnosis was confirmed through clinical examination, ultrasound, and MRI, with all patients undergoing surgical decompression. Complete recovery was achieved in all cases within a median of 12 months. However, no studies have specifically explored the use of aspiration for treating elbow synovial cysts, as existing reports focus on surgical approaches. While evidence on aspiration for elbow synovial cysts is lacking, insights can be drawn from studies on synovial cysts in other anatomical regions. For instance, Facet joint synovial cysts in the lumbar spine have been successfully treated with aspiration, with one report demonstrating sustained symptom relief over 24 months using a percutaneous two-needle approach under fluoroscopic guidance [9]. In contrast, a prospective study [10] on hip synovial cysts found that needle aspiration provided faster recovery and short-term relief but lacked durability, highlighting the variability of outcomes depending on the anatomical location. For elbow synovial cysts, evidence on recurrence rates and functional recovery remains scarce. Ultrasound-guided aspiration has been shown to be safe when careful visualization prevents injury to adjacent neural structures. In contrast, the risk of iatrogenic PIN injury during surgical resection is not well-documented. Further research is warranted to assess the long-term efficacy and safety of aspiration versus surgical excision for elbow synovial cysts.

## Data Availability

Data are contained within the main text of the manuscript.

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
