# Peer review of "A Rare Case of Elbow Synovial Cyst with Radial Nerve Compression"

_diagnostics, 2025, doi:10.3390/diagnostics15020124_

Round 1
Reviewer 1 Report
Comments and Suggestions for Authors
Thank you very much for the opportunity to evaluate this case report. In summary, the possibility of sonographic visualization of a precapsular cyst with affection of the posterior interosseous nerv is described. The authors present the possibility of sonographically guided puncture as a therapeutic option.
The case report is very detailed in its presentation and visualization. The following are brief comments:
1. abstract: no change requests
2. main part: case and method description no objections.
However, there is no critical discussion of the method. Recurrence rate with isolated puncture instead of surgical resection? Injury of the deep radial nerve.
3. illustrations: If necessary, extension of the labeling with regard to the muscles
Even if it is a case report, the method should be critically evaluated against the existing literature. Sonography not only enables differentiation with regard to lipoma, rheumatoid nodule, other inflammatory processes,it also allows a therapeutic approach.
Author Response
Comments 1: Abstract: no change requests
Response 1: Thank you so much for this positive feedback.
Comments 2: Main part: case and method description no objections. However, there is no critical discussion of the method. Recurrence rate with isolated puncture instead of surgical resection? Injury of the deep radial nerve.
Response 2: Thank you for raising this important point. We have added a discussion on the recurrence of isolated aspiration and surgical resection for synovial cysts. This addition can be found on Page 4-5, Lines 106–121."
Comments 3: Illustrations: If necessary, extension of the labeling with regard to the muscles.
Response 3: Thank you for your suggestion regarding the labeling of the muscles in the figures. We have updated Figure 1 to Figure 5 to include additional labels for the surrounding muscle structures. These updated illustrations enhance clarity and provide better anatomical context for readers.
Comments 4: Even if it is a case report, the method should be critically evaluated against the existing literature. Sonography not only enables differentiation with regard to lipoma, rheumatoid nodule, other inflammatory processes, it also allows a therapeutic approach.
Response 4: We have emphasized the significant therapeutic value of ultrasound-guided techniques, beyond its diagnostic capabilities. This includes its role in providing targeted, minimally invasive interventions. The discussion is on Page 4, Lines 91–94, 100–105.
Reviewer 2 Report
Comments and Suggestions for Authors
This is an intriguing case of a rare syndrome, PIN entrapment. Furthermore, the etiology of entrapment in the presented case is particularly uncommon, rendering the case highly distinctive. The utilization of ultrasound was pivotal in facilitating the diagnosis and in identifying the potential for compression, thereby underscoring the growing role of ultrasound in neurophysiology.
Figures are very informative but the clinical description of the muscle weakness (which muscles were affected) and the result of the neurophysiological examination are absent from the report, which would be beneficial to include.
There are no reports of previous cases of PIN cyst-related neuropathy, if any, and how they have been diagnosed and treated.
Minor comment. I don’t really understand the necessity of references [2], [3] when describing your own findings
Author Response
Comments 1: Figures are very informative, but the clinical description of the muscle weakness (which muscles were affected) and the result of the neurophysiological examination are absent from the report, which would be beneficial to include.
Response 1: Thank you for identifying this gap. We have now included a detailed description of the specific muscle weaknesses observed and the findings from the neurophysiological examination. This information has been added to Page 1, Lines 30-36.
Comments 2: There are no reports of previous cases of PIN cyst-related neuropathy, if any, and how they have been diagnosed and treated.
Response 2: Thank you for this observation. We have expanded the manuscript to review the available literature on PIN compression by synovial cysts, detailing how such cases have been diagnosed and managed. The addition can be found on Page 4, Lines 106-112.
Comments 3: Minor comment. I don’t really understand the necessity of references [2], [3] when describing your own findings.
Response 3: Thank you for bringing this to our attention. We agree that references [2] and [3] were unnecessary in the context of describing our own findings. Therefore, we have removed these references from the relevant sections in the manuscript to maintain clarity and coherence.